# Social Support and Resilience Are Protective Factors against COVID-19 Pandemic Burnout and Job Burnout among Nurses in the Post-COVID-19 Era

**DOI:** 10.3390/healthcare12070710

**Published:** 2024-03-24

**Authors:** Ioannis Moisoglou, Aglaia Katsiroumpa, Maria Malliarou, Ioanna V. Papathanasiou, Parisis Gallos, Petros Galanis

**Affiliations:** 1Faculty of Nursing, University of Thessaly, 41500 Larissa, Greece; iomoysoglou@uth.gr (I.M.); iopapathanasiou@uth.gr (I.V.P.); 2Clinical Epidemiology Laboratory, Faculty of Nursing, National and Kapodistrian University of Athens, 11527 Athens, Greece; aglaiakat@nurs.uoa.gr (A.K.); parisgallos@nurs.uoa.gr (P.G.); pegalan@nurs.uoa.gr (P.G.)

**Keywords:** social support, resilience, COVID-19, burnout, job, nurses

## Abstract

Nurses have experienced several psychological and work-related issues during the COVID-19 pandemic, including pandemic burnout and job burnout. The aim of this study was to examine the impact of social support and resilience on COVID-19 pandemic burnout and job burnout among nurses. We conducted a cross-sectional study in Greece. The study population included 963 nurses. We measured social support, resilience, COVID-19 pandemic burnout, and job burnout with the Multidimensional Scale of Perceived Social Support, Brief Resilience Scale, COVID-19 Burnout Scale, and Single-Item Burnout Measure, respectively. Nurses received high levels of social support, while their resilience was moderate. Additionally, nurses experienced moderate levels of COVID-19 burnout and job burnout. Increased social support and increased resilience were associated with reduced COVID-19 pandemic burnout. We found a negative relationship between social support and job burnout. A similar negative relationship was found between resilience and job burnout. Social support and resilience can act as protective factors against COVID-19 pandemic burnout and job burnout among nurses. Policy makers should develop and implement appropriate strategies to improve nurses’ social support and resilience since they are the backbone of healthcare systems worldwide.

## 1. Introduction

The COVID-19 pandemic has been a major challenge for citizens, health systems, and health professionals. The effort to reduce the transmission of the virus and control the pandemic involved a series of stringent measures that restricted the social and professional lives of citizens [1]. At the same time, however, there were massive hospital admissions of patients with COVID-19, with a significant proportion of them requiring intensive care in intensive care units, which was a major test of the resilience of both health systems and health professionals, particularly nurses, who have been on the front line [2]. The World Health Organization officially declared the end of the COVID-19 pandemic as a public health emergency on 5 May 2023, but by the summer of 2022, the waves of the pandemic had receded, the number of hospital admissions had significantly decreased, and health systems had returned to their pre-pandemic operation.

The response of governments worldwide to the spread of the COVID-19 pandemic included a series of stringent measures, including restrictions on citizens’ mobility and professional activities, the interruption of education, and restrictions on travel and entertainment [3]. The most stringent measure of all was quarantine. Despite the effectiveness of these measures, there have been significant effects on the well-being of citizens. There was a significant incidence of loneliness, agitation, depression, anxiety, irritability, and mental distress, as well as stress, psychological distress, and a deterioration in quality of life [4,5,6,7,8]. Also, the fear of infection of both the individuals and those close to them was a factor that had a negative impact on individuals’ mental health [9]. Even after the end of the quarantine period, people continued to experience symptoms of depression, anxiety, and acute stress disorder, according to findings from longitudinal studies [10,11]. This constant exposure to stress can lead to burnout. According to Maslach and Leiter, burnout is a psychological syndrome emerging as a prolonged response to chronic interpersonal stressors on the job, and the characteristics of its three dimensions include the loss of energy, depletion, debilitation, and fatigue; irritability, loss of idealism, and withdrawal; and reduced productivity or capability, low morale, and an inability to cope [12]. However, during the pandemic period, the incidence of burnout was not limited to workers; the general population also experienced symptoms of burnout [13,14,15]. The existing literature on nurses’ burnout is exclusively related to burnout from work. There is a lack of studies on the effect of social support and the resilience of nurses on COVID-19 pandemic burnout.

The burnout of nurses is a long-standing problem. Before the pandemic period, the incidence of burnout among nursing staff was as high as 50% of the staff [16,17]. Research on burnout has highlighted a significant number of mainly organizational factors associated with the occurrence of burnout in nurses. In particular, the work environment, characterized by a lack of staff and resources, a high workload, the inability of leadership to support staff, poor working relationships between doctors and nurses, the lack of nurses’ involvement in hospital issues, ≥12 h shifts, time pressure, high job and psychological demands, and low autonomy, constituted the primary factor that contributed to the occurrence of burnout [18,19,20,21,22]. The outcomes of burnout affect both patients and nursing staff. Concerning nurses, burnout has been associated with reduced job performance and intention to leave, and concerning patients, burnout can result in poor quality of care, poor patient safety, adverse events, patient negative experience, medication errors, infections, and patient falls [22,23]. The organizational weaknesses became even more pronounced during the pandemic, as the mass admission of patients with COVID-19 increased the workload and work intensity. This high workload, combined with nursing understaffing, lack of resources, and longer working time in quarantine areas, increased the likelihood of developing burnout [24,25]. Even now, in the post-COVID-19 era, where the impact of the pandemic on health systems has been significantly reduced and the functioning of health organizations has been normalized, burnout continues to have a very high incidence among nurses compared to other healthcare professionals [26]. This can be explained by the fact that organizational problems remain unresolved and that the impact of the pandemic on nurses is still quite recent.

In a demanding work environment, considering the highly stressful nature of nurses’ work, the support that they should receive to cope with their daily challenges is particularly crucial. One such form of support is social support, which, in general, refers to psychological or material resources that are provided to a focal individual by partners in some form of social relationship [27]. The source of social support can be family members, friends, work colleagues, a supervisor, or the organization. Social support can be provided in four forms [28]: the first is emotional support, which involves the provision of caring, empathy, love, and trust; the second one, instrumental support, concerns the provision of tangible goods, services, or concrete assistance; the third form, informational support, concerns information provided to another during a time of stress to assist nurses in problem solving; the fourth form is appraisal support, which consists of the communication of information that is relevant to self-evaluation rather than problem solving. Regardless of the source and type of social support, studies have shown their positive impact on healthcare workers. In particular, the benefits in the field of mental health are significant. Medical staff who receive social support experience lower levels of anxiety and stress and also report better self-efficacy and sleep quality [29]. Low supervisor support of healthcare professionals is associated with higher levels of secondary traumatic stress [30], and low perceived support by friends predicts burnout [31]. Nurses who perceive higher organizational and social support are more likely to report lower anxiety related to COVID-19 [32]. Also, the social support that nurses receive from family and significant others outside the family protects them from developing burnout, anxiety, and depression [33]. The support that nurses receive in the workplace from supervisors and coworkers was found to play a fundamental role in preventing the burnout syndrome [34].

One characteristic of nurses that can help them cope with their demanding work environment is resilience. Although resilience has been recognized as a trait, as a process, and as an outcome, its acceptance as a dynamic process enables the individual to develop it [35]. In the context of nursing, resilience can be defined as a “…complex and dynamic process which when present and sustained enables nurses to positively adapt to workplace stressors, avoid psychological harm and continue to provide safe, high-quality patient care” [35]. According to this definition, resilience is fundamental to maintaining the mental health of nurses in their efforts to provide high-quality care. In particular, resilience is negatively associated with burnout and secondary traumatic stress and positively associated with compassion and satisfaction [36,37], and it also plays a protective role in the development of depression [38]. Among the sources of resilience are environmental–systemic factors, which include social support [39]. Studies have shown a positive correlation between nurses’ social support and resilience [40,41].

During the COVID-19 pandemic, to date, more than 5.6 million cases have been recorded in Greece [42], with a significant number of patients being admitted to hospital and a proportion of them being hospitalized in ICUs [43]. In an effort to limit the spread of the pandemic, the government decided to implement strict restriction measures for citizens over at least four periods. At the same time, nursing staffing in Greek hospitals remains among the lowest in OECD countries [44]. Nurses, therefore, found themselves under very difficult social and working conditions. To the best of our knowledge, there is a lack of studies that investigate the relationship between social support and resilience and COVID-19 pandemic burnout and job burnout among nurses in the post-COVID-19 era. Thus, the aim of this study was to examine the impact of social support and resilience on COVID-19 pandemic burnout and job burnout in a sample of nurses.

## 2. Materials and Methods

### 2.1. Study Design

A cross-sectional study was conducted during September 2022 to examine the impact of social support and resilience on COVID-19 pandemic burnout and job burnout in the post-COVID-19 era. Our study obtained a convenience sample of nurses in Greece. We developed an online Greek version of the study questionnaire with Google Forms. We disseminated the questionnaire through Facebook, Instagram, and LinkedIn. Moreover, we sent the questionnaire to our e-mail contacts. We applied the following inclusion criteria: (a) nurses who were working in healthcare services during the COVID-19 pandemic; (b) those who understand Greek language; (c) nurses who have social media accounts. Thus, nurses who were not working in clinical settings during the COVID-19 pandemic, those who did not understand Greek, and those without social media accounts cannot participate in our study. We collected data on an anonymous and voluntary basis.

### 2.2. Measurements

The following sociodemographic characteristics of nurses were measured: gender (males of females), age (continuous variable), chronic disease (no or yes), self-perceived health status (very poor, poor, moderate, good, very good), SARS-CoV-2 infection (no or yes), providing care to COVID-19 patients (no or yes), and adverse effects because of COVID-19 vaccination (continuous variable).

The Multidimensional Scale of Perceived Social Support (MSPSS) was used to measure social support [45]. The MSPSS includes 12 items and measures support from family, friend, and significant others. Example items for the MSPSS are the following: “There is a special person who is around when I am in need”, “There is a special person with whom I can share my joys and sorrows”, and “My family really tries to help me”. The total score ranges from 1 to 7, with higher values being indicative of higher levels of social support. We used the valid Greek version of the MSPSS [46]. Cronbach’s alpha for the MSPSS in our study was 0.952.

The Brief Resilience Scale (BRS) includes six items and was used to measure resilience among our nurses [47]. Example items for the BRS are the following: “I tend to bounce back quickly after hard times”, “I have a hard time making it through stressful events”, and “It does not take me long to recover from a stressful event”. The total score ranges from 1 to 5, with higher values being indicative of higher levels of resilience. We used the valid Greek version of the BRS [48]. Cronbach’s alpha for the BRS in our study was 0.806.

The Greek version of the COVID-19 Burnout Scale (COVID-19-BS) was used to measure COVID-19 pandemic burnout [49]. The COVID-19-BS is a specific tool used to measure COVID-19-related burnout during the pandemic. Example items for the COVID-19-BS are the following: “I feel emotionally tired because of the COVID-19 pandemic”, “I feel sad because of the COVID-19 pandemic”, and “I feel angry because of the COVID-19 pandemic”. The COVID-19-BS includes 13 items with a total score from 1 to 5. Higher scores on the COVID-19-BS are indicative of higher levels of burnout. The COVID-19-BS measures emotional exhaustion, physical exhaustion, and exhaustion due to measures against the COVID-19. Cronbach’s alpha for the COVID-19-BS in our study was 0.912.

The Single-Item Burnout Measure was used to measure nurses’ burnout [50]. The Single-Item Burnout Measure has been established as a valid tool in the Greek language [51]. In this case, nurses rated their job burnout on a scale from 0 (no burnout at all) to 10 (extreme levels of burnout).

### 2.3. Ethical Considerations

Our study was conducted according to the guidelines of the Declaration of Helsinki [52]. Additionally, the Ethics Committee of the Faculty of Nursing, National and Kapodistrian University of Athens (reference number; 370, 2 September 2021), approved the study protocol. We informed nurses about the aim and the study design, and they gave their informed consent to participate. We did not collect personal data of nurses. For example, we did not ask them to provide their name, email address, or the healthcare service at which they work. Of our team, only the statistician (P.G.) has access to the data. SPSS files were protected with passwords to avoid unintended access.

### 2.4. Statistical Analysis

Our study variables were the sociodemographic characteristics of nurses (i.e., gender, age, chronic disease, self-perceived health status, SARS-CoV-2 infection, providing care to COVID-19 patients, and adverse effects because of COVID-19 vaccination), social support, resilience, COVID-19 pandemic burnout, and job burnout. Our independent variables were the sociodemographic characteristics of nurses, social support, and resilience. COVID-19 pandemic burnout and job burnout were dependent variables. Thus, our research hypotheses were the following:

**H1.** *Social support reduces COVID-19 pandemic burnout and job burnout*.

**H2.** *Resilience reduces COVID-19 pandemic burnout and job burnout*.

**H3.** *Sociodemographic characteristics affect COVID-19 pandemic burnout and job burnout*.

Sample size was estimated at 436 nurses by applying the following parameters: (a) a low effect size (f^2^ = 0.03) between social support, resilience, COVID-19 pandemic burnout, and job burnout; (b) nine independent variables; (c) 95% confidence level; and (d) 5% margin of error.

Categorical variables are presented with numbers and percentages. Also, we present continuous variables with mean, standard deviation (SD), median, minimum value, and maximum value. We used the Kolmogorov–Smirnov test and Q–Q plots to assess the distribution of continuous variables. First, we performed univariate linear regression analysis between each independent variable and COVID-19 pandemic burnout and job burnout. Then, we created two multivariable linear regression models with COVID-19 pandemic burnout and job burnout as the dependent variables to eliminate confounding causes via sociodemographic characteristics. We present unadjusted and adjusted coefficients beta, 95% confidence intervals (CI), *p*-values, and coefficients of determination (R^2^). Questionnaires with more than 20% missing values were removed from the final analysis. We considered *p*-values < 0.05 as statistically significant. Similarly, confidence intervals that included zero were indicative of statistically significant relationships. We used the IBM SPSS 21.0 (IBM Corp. Released 2012. IBM SPSS Statistics for Windows, Version 21.0. Armonk, NY, USA: IBM Corp.) for the analysis.

## 3. Results

### 3.1. Sociodemographic Characteristics

Our study population included 963 nurses. The majority of our nurses were females (88.4%). The mean age of the nurses was 37.9 (SD = 9.6). Among our sample, 25% had a chronic disease, while 71.8% have been infected by SARS-CoV-2 during the pandemic. Most of the nurses (78.3%) self-assessed their health status as good/very good, while 7.3% considered their health status to be moderate, and 4.4% considered it to be very poor/poor. During the pandemic, 64.1% of the nurses provided healthcare to COVID-19 patients. We found that all the continuous variables followed a normal distribution. Table 1 shows the sociodemographic characteristics of the nurses.

### 3.2. Study Scales

Descriptive statistics for the study scales are shown in Table 2. Nurses received high levels of social support (mean = 6.0, SD = 1.2), while their resilience was moderate (mean = 3.5, SD = 0.7). Median values for social support and resilience were 6.4 and 3.5, respectively. Additionally, nurses experienced moderate levels of COVID-19 burnout (mean = 3.2, SD = 1.0) and job burnout during the pandemic (mean = 6.4, SD = 2.6). Median values for COVID-19 burnout and job burnout during the pandemic were 3.2 and 7.0, respectively. Scores on all study scales followed a normal distribution.

### 3.3. Impact on COVID-19 Pandemic Burnout

After eliminating confounders, we found that increased social support (adjusted beta = −0.075, 95% CI = −0.125 to −0.024, *p*-value = 0.004) and increased resilience (adjusted beta = −0.399, 95% CI = −0.491 to −0.308, *p*-value < 0.001) were associated with reduced COVID-19 pandemic burnout. Moreover, females (adjusted beta = 0.201, 95% CI = 0.024 to 0.378, *p*-value = 0.026) and nurses who provided care to COVID-19 patients (adjusted beta = 0.147, 95% CI = 0.031 to 0.263, *p*-value = 0.013) experienced higher levels of COVID-19 pandemic burnout. We found a positive relationship between the level of adverse effects because of COVID-19 vaccination and COVID-19 pandemic burnout (adjusted beta = 0.083, 95% CI = 0.061 to 0.105, *p*-value < 0.001). We present our linear regression analysis with COVID-19 pandemic burnout as the dependent variable in Table 3.

### 3.4. Impact on Job Burnout

Table 4 shows the results from our linear regression analysis with job burnout as the dependent variable. After eliminating confounders, we found a negative relationship between social support and job burnout (adjusted beta = −0.263, 95% CI = −0.405 to −0.121, *p*-value < 0.001). A similar negative relationship was found between resilience and job burnout (adjusted beta = −0.529, 95% CI = −0.785 to −0.272, *p*-value < 0.001). Moreover, levels of job burnout were higher among nurses with a chronic disease (adjusted beta = 0.743, 95% CI = 0.367 to 1.119, *p*-value < 0.001) and those who provided care to COVID-19 patients (adjusted beta = 0.677, 95% CI = 0.353 to 1.002, *p*-value < 0.001). Additionally, we found a positive relationship between the level of adverse effects because of COVID-19 vaccination and job burnout (adjusted beta = 0.148, 95% CI = 0.086 to 0.210, *p*-value < 0.001).

## 4. Discussion

The present study highlighted the protective role of social support and resilience on the occurrence of nurses’ work burnout and that related to the COVID-19 pandemic. This study was one of the first studies that investigated nurses’ burnout due to the COVID-19 pandemic and its association with social support and resilience, as there is a lack of studies on the relevant topics. Also, studies in the literature are limited to workplace burnout only, and though they explore the relationship of social support and resilience with mental health, they do not do so specifically with pandemic burnout.

According to the findings of the present study, nurses experienced moderate levels of COVID-19 burnout. The social and professional conditions created during the pandemic period provided fertile ground for the development of pandemic burnout. Since, as we mentioned, there is a deficiency in the relevant literature regarding pandemic burnout, we posit that the conditions created during the pandemic probably influenced our results regarding the occurrence of pandemic burnout; specifically, the adoption of strict measures, such as quarantine, in response to the spread of the pandemic, along with the fear of infection, affected the mental health of the general population together with that of nurses [53,54,55]. Nursing staff were not excluded from the implementation of the pandemic protection measures, and in particular the quarantine, although they were allowed to move to and from work. The nurses, therefore, on the one hand, experienced an extremely difficult working period with overtime work during the pandemic, and on the other hand, on returning home, there was no option for social activity. According to the Our World in Data database, the COVID-19 stringency index is a composite measure based on nine response indicators including school closures, workplace closures, and travel bans, rescaled to a value from 0 to 100 (100 = strictest) [56]. In Greece, between March 2020 and February 2022, there were four periods when the value of this index exceeded 80, meaning that the measures in these periods were very strict. Therefore, prolonged periods of strict pandemic control measures restricting social activities, the fear of infection, and at the same time, the workload of nurses created a mixture of stressful conditions that favored the development of COVID-19 pandemic burnout. Also, among health professionals, nurses were found to have the highest rates of work-related burnout both during and after the pandemic [25,26]. These high rates of work-related burnout may also have influenced burnout from the pandemic and may explain the findings of this study. However, there is a need for more studies on pandemic burnout and its determinants.

Social support and resilience are two important predictors of prosocial behaviors, as these behaviors refers to a wide range of actions such as helping, sharing, comforting, and cooperating, and are key elements for the proper functioning of society, especially in the face of a crisis such as COVID-19 [57]. According to the findings of the present study, nurses experienced high levels of social support and moderate levels of resilience, which, in turn, had a significant negative effect on both pandemic and work-related burnout. Our findings are consistent with those of the literature, as during the pandemic period, social support and resilience were identified as protective factors with respect to the mental health of the general population, separated into different age groups and professional activities [58,59,60]. The populations’ benefits from social support and resilience remain significant even after the impact of the pandemic [61], including for patients with post-COVID-19 syndrome, who were found to experience better mental health and quality of life due to social support and resilience [62].

Our findings regarding social support and resilience are consistent with those of the literature concerning nursing staff. The protective effect of social support and resilience on nurses, who were a professional group with a very high incidence of burnout, was significant. In particular, when nurses do not receive adequate support from their supervisor, they are more likely to develop burnout [63]. Moral resilience can be an essential protective factor against high levels of job burnout, quiet quitting, and turnover intention among nurses [64]. Organizational support is also an important form of support for nurses which is associated with burnout. When nurses receive recognition and their organization values their contribution and cares about their well-being—which is called perceived organizational support [65]—and, at the same time, they have high resilience, then the likelihood of burnout is reduced [37,66]. The most important forms of organizational support that healthcare organizations should ensure for nurses include organizational rewards, favorable job conditions, assistance to nurses in performing tasks efficiently and managing stressful situations, and support from their supervisor [67]. Social support and resilience seem to have a multidimensional positive effect on nurses as they protect them not only from burnout, as shown in this study, but also from the negative aspects of their profession.

Social support also has an impact on resilience; that is, the more social support nurses receive, the better their resilience [68,69]. The resilience of nurses can also be fostered by the supervising nurse. The actions through which nurses’ resilience is cultivated by supervisors include facilitating social connections, promoting positivity, capitalizing on nurses’ strengths, nurturing nurses’ growth, encouraging nurses’ self-care, fostering mindfulness practice, and conveying altruism [70]. Therefore, enhancing nurses’ social support (including, of course, organizational support) and fostering resilience can be important interventions that will have a positive impact on nurses’ well-being. An important feature of resilience is the fact that it can be cultivated through educational interventions; studies have shown the effectiveness of training programs in reducing stress, improving general health, enhancing resilience, and confronting adversity [71,72,73].

The present study highlighted nurses’ care of patients with COVID-19 as a contributing factor to their pandemic and job burnout, as well as the existence of chronic disease among nurses as a predictive factor of job burnout. Nurses working in the nursing departments for patients with COVID-19 experienced extremely difficult conditions in and outside of their workplace. Outside the hospital and in the community, they experienced harsh quarantine measures with restrictions on their movement, were isolated in a different room of the house from the rest of the family, or were moved to another house on their own to avoid the transmission of COVID to their family. They experienced stigma as their community members avoided socializing with them [74]. However, the support that nurses received, mainly from family and friends, helped them to attain better mental health and, in particular, a lower likelihood of fear, depression, anxiety, and stress [75]. The working conditions in the COVID-19 wards and ICUs were extremely stressful and burdensome for the nurses. High workload, overtime, and negative ratings with respect to workplace relations, organizational support, organizational preparedness, psychological support, workplace safety, and access to supplies and resources characterized their working environment. These conditions led to the development of mental health issues such as burnout, depression, anxiety, and post-traumatic stress disorder [25,76,77,78]. The existence of chronic disease among nurses increased their stress and their fear of contracting a possible severe case of COVID-19. During the pandemic, patients with chronic disease were identified as vulnerable groups as they were more likely to experience severe disease, hospital admission, and/or mortality from COVID-19 than those without chronic disease [79,80]. Studies have shown that the greater the nurses’ fear of COVID-19 infection, the higher their degree of burnout, which is consistent with the findings of our study [81,82]. Support and attention from the organization, alongside building resilience, can help nurses to manage their abundant physical and emotional stress while treating COVID-19 patients, which arose from risk of infection [83]. 

Our study had several limitations. First, we conducted a cross-sectional study to explore the impact of social support and resilience on COVID-19 pandemic burnout and job burnout in a sample of nurses in Greece. Although we eliminated several confounders, the cross-sectional nature of our study did not allow us to infer causal relationships between social support and resilience and COVID-19 pandemic burnout and job burnout. Second, future studies should eliminate more confounders to establish a more valid relationship between social support and resilience and COVID-19 pandemic burnout and job burnout among nurses. Additionally, we used self-reported questionnaires to measure social support, resilience, COVID-19 pandemic burnout, and job burnout. Although these scales are valid, information bias was probable in our study. Additionally, we used a convenience sample with an unknown response rate. Thus, selection bias can arise in our study and one should generalize our results with caution. Further research with random and more representative samples of nurses could add valuable information. Studies in different countries and clinical settings are necessary in order to improve our knowledge. Moreover, longitudinal studies will offer substantial knowledge on the issue by monitoring changes in levels of social support, resilience, and burnout among nurses.

## 5. Conclusions

The COVID-19 pandemic, containment measures, and fear of infection have taken a toll on the mental health of the general population. Nurses were not excluded from this burden. According to our study, nurses experienced moderate levels of COVID-19 burnout and job burnout in this post-COVID-19 period. Important protective factors against burnout—both pandemic and job burnout—were highlighted, namely, social support and resilience. Alongside the support that nurses can receive from family and friends, organizational support is also essential. Although the impact of the pandemic on health systems has subsided, the working environment for nurses continues to be demanding, putting a burden on nurses’ well-being. Therefore, the need to support nurses is ongoing. Recognition of nurses’ work, rewarding them, and ensuring they receive all the necessary resources and daily support from supervisors represent some of the most significant forms of organizational support, which can reduce burnout and foster resilience. As resilience is an important skill for nurses which can be cultivated through training programs, nurses’ managers, stakeholders, and organizations should adopt appropriate interventions to improve resilience and, at the same time, enhance the social support of nurses. In this context, negative consequences such as burnout may be reduced.

## Figures and Tables

**Table 1 healthcare-12-00710-t001:** Sociodemographic characteristics of the nurses (N = 963).

Variables	N	%
Gender		
Males	112	11.6
Females	851	88.4
Age (years) ^a^	37.9	9.6
Chronic disease		
No	722	75.0
Yes	241	25.0
Self-perceived health status		
Very poor	26	2.7
Poor	16	1.7
Moderate	70	7.3
Good	580	60.2
Very good	271	28.1
SARS-CoV-2 infection		
No	272	28.2
Yes	691	71.8
Providing care to COVID-19 patients		
No	346	35.9
Yes	617	64.1
Adverse effects because of COVID-19 vaccination ^a^	3.1	2.6

^a^ mean, standard deviation.

**Table 2 healthcare-12-00710-t002:** Descriptive statistics for the study scales.

Scale	Mean	Standard Deviation	Median	Minimum Value	Maximum Value
Social support	6.0	1.2	6.4	1.1	7.0
Resilience	3.5	0.7	3.5	1.0	5.0
COVID-19 burnout	3.2	1.0	3.2	1.1	5.0
Job burnout	6.4	2.6	7.0	0	10

**Table 3 healthcare-12-00710-t003:** Univariate and multivariable linear regression analysis with COVID-19 pandemic burnout as the dependent variable.

Independent Variables	Univariate Model		Multivariable Model		Hypothesis Number	Accepted or Rejected
	Unadjusted Coefficient Beta (95% CI)	*p*-Value	Adjusted Coefficient Beta (95% CI) ^a^	*p*-Value		
Females vs. males	0.334 (0.146 to 0.522)	0.001	0.201 (0.024 to 0.378)	**0.026**	3	Accepted
Age (years)	−0.001 (−0.008 to 0.005)	0.643	−0.002 (−0.008 to 0.004)	0.433	3	Rejected
Chronic disease (yes vs. no)	0.154 (0.014 to 0.294)	0.031	0.117 (−0.017 to 0.251)	0.088	3	Rejected
Self-perceived health status	−0.126 (−0.201 to −0.052)	0.001	−0.013 (−0.085 to 0.059)	0.724	3	Rejected
SARS-CoV-2 infection (yes vs. no)	0.015 (−0.120 to 0.149)	0.832	0.042 (−0.084 to 0.169)	0.511	3	Rejected
Providing care to COVID-19 patients (yes vs. no)	0.128 (0.002 to 0.254)	0.046	0.147 (0.031 to 0.263)	**0.013**	3	Accepted
Adverse effects because of COVID-19 vaccination	0.101 (0.079 to 0.124)	<0.001	0.083 (0.061 to 0.105)	**<0.001**	3	Accepted
Social support	−0.149 (−0.200 to −0.099)	<0.001	−0.075 (−0.125 to −0.024)	**0.004**	1	Accepted
Resilience	−0.492 (−0.580 to −0.405)	<0.001	−0.399 (−0.491 to −0.308)	**<0.001**	2	Accepted

Bold *p*-values indicate statistically significant associations in the multivariable model. CI: confidence interval. ^a^ *p*-value for ANOVA < 0.001. R^2^ for the final multivariable model was 17.9%.

**Table 4 healthcare-12-00710-t004:** Univariate and multivariable linear regression analysis with job burnout as the dependent variable.

Independent Variables	Univariate Model		Multivariable Model		Hypothesis Number	Rejected or Accepted
	Unadjusted Coefficient Beta (95% CI)	*p*-Value	Adjusted Coefficient Beta (95% CI) ^a^	*p*-Value		
Females vs. males	0.184 (−0.325 to 0.692)	0.478	0.029 (−0.466 to 0.524)	0.909	3	Rejected
Age (years)	0.013 (−0.004 to 0.030)	0.139	−0.0004 (−0.017 to 0.016)	0.959	3	Rejected
Chronic disease (yes vs. no)	0.960 (0.588 to 1.331)	<0.001	0.743 (0.367 to 1.119)	**<0.001**	3	Accepted
Self-perceived health status	−0.393 (−0.593 to −0.193)	<0.001	−0.117 (−0.319 to 0.085)	0.256	3	Rejected
SARS-CoV-2 infection (yes vs. no)	−0.258 (−0.620 to 0.104)	0.162	−0.121 (−0.476 to 0.235)	0.505	3	Rejected
Providing care to COVID-19 patients (yes vs. no)	0.728 (0.392 to 1.065)	<0.001	0.677 (0.353 to 1.002)	**<0.001**	3	Accepted
Adverse effects because of COVID-19 vaccination	0.170 (0.108 to 0.232)	<0.001	0.148 (0.086 to 0.210)	**<0.001**	3	Accepted
Social support	−0.446 (−0.582 to −0.311)	<0.001	−0.263 (−0.405 to −0.121)	**<0.001**	1	Accepted
Resilience	−0.759 (−1.005 to −0.514)	<0.001	−0.529 (−0.785 to −0.272)	**<0.001**	2	Accepted

Bold *p*-values indicate statistically significant associations in the multivariable model. CI: confidence interval. ^a^ *p*-value for ANOVA < 0.001. R^2^ for the final multivariable model was 11.0%.

## Data Availability

The data presented in this study are available on request from the corresponding author.

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
