# Peer review of "Social Support and Resilience Are Protective Factors against COVID-19 Pandemic Burnout and Job Burnout among Nurses in the Post-COVID-19 Era"

_healthcare, 2024, doi:10.3390/healthcare12070710_

Round 1

Reviewer 1 Report

Comments and Suggestions for Authors

Unfortunately, besides interesting, and important topics – Social support and resilience are protective factors against COVID-19 pandemic burnout and job burnout among nurses in the post-COVID-19 era - the manuscript has numerous flaws.

The manuscript is too extensive and repeats well-known facts about the problem under research (social support, resilience, burnout syndrome). This fact is supported by the overlap percentage of 62% on the iThenticate report.

Also, the authors point out several times that their study is the only one that investigates the problems mentioned above, and they have 75 references, indicating how much research has already been published.

In addition, the authors use an informal style in the methodology part, which does not correspond to a scientific manuscript. Each paragraph in the methodology begins with We measured, performed, and described.

In the discussion, the authors do not analyze their study results but consider the research problem in general.

Author Response

Dear Reviewer 1

Thank you very much for the peer review of the paper “Social support and resilience are protective factors against COVID-19 pandemic burnout and job burnout among nurses in the post-COVID-19 era” and your comments, which have improved the quality of the manuscript.

We have addressed all the comments (highlighted in yellow) in the revised text. Please, find below an item-by-item answer to your comments.

Hoping the revised manuscript fulfils the journal’s standards, we thank you for your courtesy.

We are looking forward to your response.

Yours sincerely,

The authors

Reviewer 2 Report

Comments and Suggestions for Authors

I reviewed the manuscript entitled “Social support and resilience are protective factors against COVID-19 pandemic burnout and job burnout among nurses in the post-COVID-19 era”. This study discussed crucial issues related to pre-COVID-19. However, there are still many improvements that need to be considered by the authors to enhance the quality of this study. I hope the author does not feel offended by my comments, which were only raised to help them.

·       The abstract is good. However, I suggest you remove the results that appeared in it. It is better to write about your main results without mentioning the adjusted beta number.

·       In the introduction, the first paragraph provided facts without references.

·       In Line 58: I don’t think it is right to say that this is the first study; rather than that, you can write there is a lack of studies on this topic in a specific industry or country. Also, the same in line 120.

·       The introduction is well written. However, the problem statement among nurses in Greece is not clear. Why did you choose to conduct your study in this specific sector? In addition, based on your title and variables, I think there should be many objectives, not only one.

·       In the methodology, please add information about the population and also how you identified your sampling size. Moreover, it is better to add the items of a questionnaire that you used in your study.

·       As long as you are going through the cross-sectional study, there must be a hypothesis to test. Please state it by drawing the framework that represents the variables of your study.

·       In statistical analysis, I would suggest rewriting the whole paragraph to make it clearer. It is better to start writing about the variables of your study and conclude by drawing your study framework. Then, you can write about the analysis method that you use in your study. Also, in this section, you can only write about your analysis method, so results about the normalization are better in the results section.

·       In the results section, authors should add sub-titles; I think that will help future readers avoid confusion.

·       In Table 3, please add the hypothesis number column and the outcomes column, for example, whether the hypothesis was accepted or rejected.

·       In the discussion, the author should discuss the results objective by objective or hypothesis by hypothesis.

·       Finally, in the conclusion, the practical and theoretical contributions are missing.

Author Response

Dear Reviewer 2,

Thank you very much for the peer review of the paper “Social support and resilience are protective factors against COVID-19 pandemic burnout and job burnout among nurses in the post-COVID-19 era” and your comments, which have improved the quality of the manuscript.

We have addressed all the comments (highlighted in yellow) in the revised text. Please, find below an item-by-item answer to your comments.

Hoping the revised manuscript fulfils the journal’s standards, we thank you for your courtesy.

We are looking forward to your response.

Yours sincerely,

The authors

Reviewer 3 Report

Comments and Suggestions for Authors

I would like to thank the editors and authors for the opportunity to review the article "Social support and resilience are protective factors against COVID-19 pandemic burnout and job burnout among nurses in the post-COVID-19 era"

The selected theme is extremely relevant, aiming to understand the role of social support and resilience as elements of protection against pandemic exhaustion resulting from COVID-19, as well as professional burnout among nurses in the post-COVID-19 era. The proposal is to provide guidelines that guide policy makers in the development and implementation of strategies designed to improve the social support and resilience of nurses, consequently promoting the health of this professional category.

The introduction provides an overview of the scientific foundations and rationale for the research being reported.

I will now offer my contributions or suggestions to improve the manuscript:

2. Materials and Methods

The text in lines 127 to 130 seems more appropriate in the introduction than in the method. Eligibility criteria, sources and methods of participant selection, as well as determination of study size are not clearly delineated. Exclusion criteria are not evident.

It is crucial to clarify how the anonymity, confidentiality and privacy of participants were guaranteed. Authors must provide information about who had access to the collected data, how this data was protected, and what the plan is for future destruction of this information.

2.4. Statistical analysis

Provide a detailed explanation of the sample size determination process and clarify the approach taken to deal with missing data.

4. Discussion

In the discussion section, it would be enriching to explore how the data obtained could provide practical guidance for policy makers during the process of designing and implementing strategies to improve nurses' social support and resilience. An approach is recommended that details the specific nature of the proposed strategies, offering concrete suggestions on how they can be effectively implemented. This level of depth would allow for a more tangible and applicable analysis of the implications of the results, contributing significantly to the formulation of effective policies in the area in question and, therefore, generating substantial practical implications.

References

In general terms, I can say that the article presents specific references of interest and the majority are less than 5 years old (88%).

Final decision:

The manuscript needs small changes.

I hope that my contributions serve to improve this article and the study you propose.

Thank you very much.

Author Response

Dear Reviewer 3,

Thank you very much for the peer review of the paper “Social support and resilience are protective factors against COVID-19 pandemic burnout and job burnout among nurses in the post-COVID-19 era” and your comments, which have improved the quality of the manuscript.

We have addressed all the comments (highlighted in yellow) in the revised text. Please, find below an item-by-item answer to your comments.

Hoping the revised manuscript fulfils the journal’s standards, we thank you for your courtesy.

We are looking forward to your response.

Yours sincerely,

The authors

Round 2

Reviewer 1 Report

Comments and Suggestions for Authors

Although the authors did not accept all the suggestions given, the revised version of the manuscript tilted Social support and resilience as protective factors against COVID-19 pandemic burnout and job burnout among nurses in the post-COVID-19 era is now easy to read, and all parts of the manuscript are better presented, which has improved the overall quality of the manuscript.

Considering those mentioned earlier and the importance of the topic the authors examined, I recommend that the manuscript be accepted.